# Maternal body mass index in early pregnancy is associated with overweight and obesity in children up to 16 years of age

Lisa Önnestam[1], Orsolya Haahr Vad[2,3], Tobias Andersson[4,5], Åsa Jolesjö[1], Jenny Sandegård[1], Kristina Bengtsson Boström[4,5]*

1 Närhälsan Ågårdsskogens Health Care Centre, Lidköping, Sweden, 2 Närhälsan Vårgårda Health Care Centre, Vårgårda, Sweden, 3 Närhälsan Nossebro Health Care Centre, Nossebro, Sweden, 4 R&D Centre Skaraborg Primary Care, Skövde, Sweden, 5 School of Public Health and Community Medicine, Institute of Medicine, Sahlgrenska Academy, University of Gothenburg, Göteborg, Sweden

* kristina.bengtsson@allmed.gu.se

**Citation:** Önnestam L, Vad OH, Andersson T, Jolesjö Å, Sandegård J, Bengtsson Boström K (2022) Maternal body mass index in early pregnancy is associated with overweight and obesity in children up to 16 years of age. PLoS ONE 17(10): e0275542. https://doi.org/10.1371/journal.pone.0275542

**Data Availability Statement:** Data cannot be shared publicly because of ethical and legal restrictions from the Swedish authorities as the

## Abstract

### Aims

Childhood obesity is an increasing public health problem. The aim of this study was to investigate the correlation between maternal body mass index in early pregnancy and body mass index in children up to the age of 16 years, and to estimate the prevalence of childhood overweight and obesity in a rural municipality in Sweden.

### Methods

The study population comprised 312 pregnant women who attended the antenatal clinics in Lidköping during the year 1999 and their 319 children. Data on body mass index from antenatal clinics, child health care centres and school health care were used in linear and multinomial logistic regressions adjusted for maternal age, smoking status, and parity.

### Results

Overweight or obesity were found in 23.0% of 16-year-olds. The correlation between maternal and child body mass index at all studied ages was positive and significant. Body mass index in 16-year-old boys showed the strongest correlation with maternal body mass index (adjusted r-square = 0.31). The adjusted relative-risk ratio for 16-year old children to be classified as obese as compared to normal weight, per 1 unit increase in maternal body mass index was 1.46 (95% confidence interval 1.29–1.65, p<0.001). Among adolescents with obesity, 37.6% had been overweight or obese at 4 years of age.

### Conclusions

This study confirms the correlation between maternal and child body mass index and that obesity can be established early in childhood. Further, we showed a high prevalence of overweight and obesity in children, especially in boys, in a Swedish rural municipality. This

data contain sensitive and potentially identifying patient information. Data may however be available for researchers who meet the criteria for access to confidential data, upon reasonable request to the authors and with permission from the Swedish Ethical Review Authority (https://etikprovningsmyndigheten.se, registrator@etikprovning.se).

**Funding:** The work was supported by the Skaraborg research and development council (grant: VGFOUSKB-869761).

**Competing interests:** The authors have declared that no competing interests exist.

suggests a need for early intervention in the preventive work of childhood obesity, preferably starting at the antenatal clinic and in child health care centres.

## Introduction

Obesity is a major and increasing public health problem among children all over the world and it results in huge societal costs [1–4]. The prevalence of overweight and obesity is increasing in both high-income and low-income countries, even though the increase seems to level out during later years in western countries such as Sweden where the national prevalence of overweight, including obesity, was 12.8% for girls and 16.0% for boys 12 years old in 2016 [5]. The prevalence has been shown to be lower among children who live in urban areas and/or have mothers with higher education [2, 3]. The Public Health Agency of Sweden reported overweight and obesity in 21% of Swedish high school students 16–19 years old in 2020. European studies have shown a correlation between body mass index (BMI) in early childhood and BMI later in adolescence and early adulthood and that adolescents with obesity were obese from the age of 5 years with accelerating BMI between 2 and 6 years of age [4, 6]. Thus, increasing BMI in children predicts overweight and obesity in adolescence and is therefore a target for preventive measures.

Childhood obesity may increase the risk of asthma, psychological illness and sleep disturbances and long-term effects such as infertility, diabetes type 2, hypertension, ischemic heart disease, stroke, and arthritis [7]. Even in early adulthood, individuals with obesity in childhood have an increased risk of death [8].

Twin and adoption studies have suggested that genetic factors contribute up to 70% of BMI, but environmental factors contribute as well [9]. In these studies, the genetic impact was lowest in mid-childhood and increased in the teens. Numerous studies have shown an association between maternal overweight and obesity versus overweight and obesity among their children in different ages, and in most studies maternal BMI (mBMI) is positively associated with child BMI [10, 11]. However, in many of these studies children were only followed until early or mid-childhood. Several studies have shown a stronger correlation between mBMI and child BMI, compared to paternal BMI and child BMI [12, 13], suggesting that in addition to genetic influences, the in utero environment may have an impact on the child's growth later in life [14]. The impact of adverse uterine environment factors on metabolic traits later in life differs between the sexes as described in a newly published review [15]. However, while some observational studies indicate that boys may have a higher risk of developing obesity following maternal over-nutrition [16] the opposite has also been found [17].

As there are few longitudinal studies that investigate the correlation of BMI between mother and child, the aim of this study was to explore the correlation between the mother´s BMI in early pregnancy and BMI in boys and girls up to the age of 16 years. The aim was also to investigate the prevalence of overweight and obesity and the development of BMI from early childhood into adolescence.

## Material and methods

### Setting

In Sweden, all pregnant women are offered free healthcare and monitoring at antenatal clinic during pregnancy and the children are offered free health care at Child Health Care (CHC) centres that focus on prevention of diseases and injuries. Growth is regularly measured up to

the age of 6 years by specialized nurses and the children are offered vaccinations. About 99% of children in Sweden aged 0–6 years participate in the program [18]. Thereafter, school healthcare with specialized nurses offers regular health visits up to 12th grade.

## Study population

The municipality Lidköping, in southwestern Sweden had approximately 38 000 inhabitants in 1999 and consists of a small urban area surrounded by rural areas. The study population comprised all pregnant women who attended the antenatal clinics during the year 1999 and their children born 1999–2000. From medical records of the first visit in early pregnancy data on age of the pregnant woman, gestational week, parity, height, weight, BMI, and smoking habits were retrieved. The children were identified by using the unique personal identity number assigned to each Swedish resident [19] in the medical records from the CHC centres. Information about date of birth, sex, gestational age, birth height, birth weight, body height and weight at 1- and 4-year routine check-ups were collected. Data from mother and child pairs were linked in a register. The register was completed with data on body weight and height at the ages of 7, 10, 13 and 16 years as well as date of examination at school healthcare. If data from multiple visits were available for a child, we chose the visit closest to 7, 10, 13 and 16 years of age. A population register was used to locate the children's home address to find the schools of children who had moved. The study was approved by the Regional Ethics Review Board in Gothenburg (reference 935–18). Informed consent was not retrieved from the studied individuals since we are reporting results from retrospective data from registers. The need for consent was waived by the ethics committee.

## Overweight and obesity

BMI was calculated for both mother and child using weight and height registered in the medical records from the antenatal clinics, CHC centres and school health care. For the mother underweight was defined according to the World Health Organisation as BMI $<18.5$ kg/m$^2$, normal weight as BMI 18.5–24.9 kg/m$^2$, overweight as BMI 25.0–29.9 kg/m$^2$, and obesity as BMI $\geq$30.0 kg/m$^2$ [20]. Mothers with underweight were categorized as normal weight in the analyses due to the small number (n = 2). Because BMI in children varies naturally with age, overweight and obesity were assessed using iso-BMI with cut-offs for overweight and obesity according to the classification by Cole et al. [21] on exact age at time of examination. Likewise, underweight was classified using the iso-BMI cut-off scale [22]. Iso-BMI is useful from 2 years of age and therefore overweight and obesity was not assessed at birth and at the age of 1, instead body weight was used to compare with mBMI.

## Statistical methods

Descriptive statistics were used to describe the study population and the prevalence of overweight and obesity in the pregnant mothers and their children in different age groups. Chi2-test was used to explore sex differences in prevalence of overweight and obesity, as compared to under- or normal weight in each child age group. Difference in median weight and BMI according to sex was tested by the Mann-Whitney test. Univariate linear regression was used to analyse the association between mBMI in early pregnancy and child BMI at 4, 7, 10, 13 and 16 years of age, for the whole study population and for boys and girls separately. Up to 1 year of age body weight was used in the linear regression to analyse association with mBMI. In an additional linear regression model, analyses were adjusted for maternal age, parity and smoking. Multinomial logistic regression was used to calculate relative-risk ratios with 95% confidence intervals (CI) for 4, 7, 10, 13, and 16 year old children to be underweight, overweight or

obese as compared to normal weight per 1 unit increase in mBMI, and additionally per 5 units increase in mBMI to correlate to a 1 step increase in mBMI-category from normal weight to overweight to obesity. In an adjusted model, maternal age, parity and smoking were used as covariates. As some mothers had multiple births, robust standard errors were used in the linear and multinomial logistic regression models. We also divided the 16 year-olds into 4 subgroups according to weight class (underweight, normal weight, overweight or obesity) and determined the percentage of children with underweight, normal weight, overweight or obesity at 4, 7, 10 and 13 years for each subgroup to track development of BMI during childhood. All analyses were performed with the use of Stata version 17.0 (Stata Corp., College station, TX, US). All tests were two-tailed and conducted at 0.05 significance level.

## Results

### Clinical characteristics of the mothers

The number of pregnancies registered at the antenatal clinics in Lidköping during 1999 were 371, Fig 1. The study population finally consisted of 312 mothers and their 319 children. The

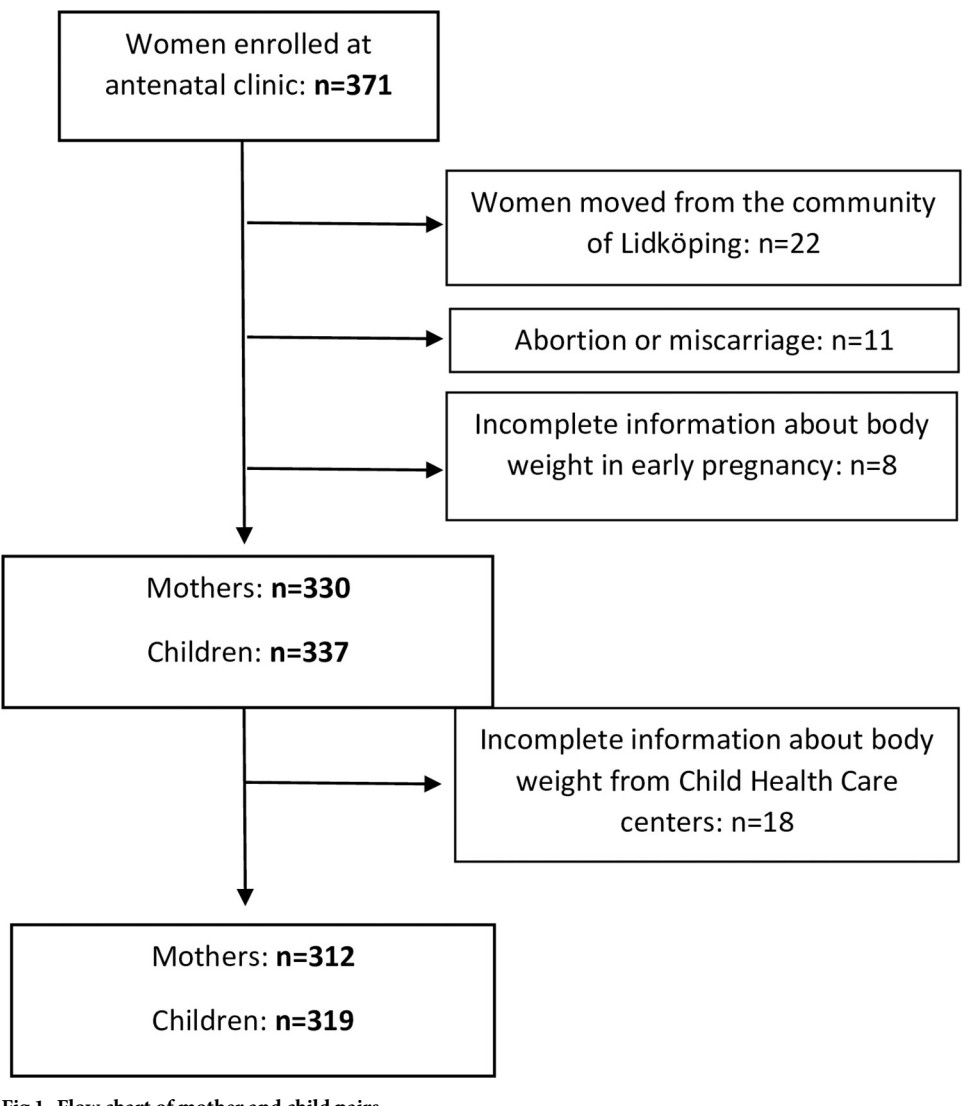

**Fig 1. Flow chart of mother and child pairs.**

women were 28.8±5.0 years and the first visit at the clinic was at 11.5±2.0 weeks of pregnancy, Table 1. Parity varied between 0–7 with median of 1 (interquartile range 2). BMI in mothers was 24.7±4.1 kg/m². The prevalence of overweight and obesity was 25.3% (n = 79) and 11.2% (n = 35) respectively.

### Clinical characteristics of the children

New-born boys (n = 168) were slightly heavier than girls (n = 151) (median weight 3.77 kg versus 3.49 kg, p = 0.0012) and this difference remained until 1 year of age (median weight 10.52 kg versus 9.90 kg, p<0.001). Median BMI for boys and girls at 4, 7, 10, 13, and 16 years of age showed no sex differences and BMI increased with increasing age for both sexes. At 4 years of age, the proportion of girls classified as either having overweight or obesity as compared to under- or normal weight was significantly larger than among boys (24.5% versus 15.0%, p = 0.032), Fig 2. At 7, 10 and 13 years there was no such difference. From 13 to 16 years of age the prevalence of obesity increased for both boys and girls to 8.3% and 3.1%, respectively. At the age of 16 years the proportion of boys and girls with either overweight or obesity were 30.5% versus 14.6% (p = 0.002). In all 16-year-old children (n = 274) 23.0% had either overweight or obesity.

### Association between maternal BMI in early pregnancy and child weight/BMI

There was a positive, significant association between mBMI in early pregnancy and the child's body weight at birth and at 1 year. Overall, there was a positive significant association between mBMI and child BMI from 4 to 16 years for the whole study group, Table 2. The strongest association was found between mBMI and BMI for 16-year-old boys where one BMI unit increase in mBMI was associated with a 0.56 kg/m² (95% CI 0.37–0.76, p<0.001) increase in child BMI in the adjusted model.

### Maternal BMI and risk of child overweight or obesity

Relative-risk ratios for children 4, 7, 10, 13 and 16 years of age to be classified as overweight or obese as compared to normal weight, per 1 unit increase in mBMI is presented in Table 3, and per 5 unit increase in mBMI in S1 Table. For example, the relative-risk ratio for 16-year-olds to be classified as obese as compared to normal weight in the adjusted model was 1,46 (95% CI 1.29–1.65) if mBMI increased with 1 unit.

**Table 1. Characteristics of women at time of enrollment at the antenatal clinic in Lidköping 1999 presented in body mass index (BMI) classification groups.**

|  | Total (n = 312) | Normal weight* (n = 198) | Overweight (n = 79) | Obesity (n = 35) |
|---|---|---|---|---|
| Age (years) | 28.8 (5.0) | 28.6 (5.0) | 29.4 (5.1) | 29.2 (4.9) |
| Height (cm) | 166.4 (5.6) | 166.7 (5.7) | 165.4 (5.4) | 166.9 (5.4) |
| Weight (kg) | 68.5 (12.2) | 61.9 (6.5) | 74.3 (5.8) | 92.8 (10.5) |
| BMI (kg/m2) | 24.7 (4.1) | 22.3 (1.7) | 27.1 (1.2) | 33.2 (2.6) |
| Parity | 1.0 (2.0) | 1.0 (1.0) | 1.0 (2.0) | 1.0 (2.0) |
| Gestational week | 11.5 (2.0) | 11.4 (1.8) | 11.5 (2.1) | 11.9 (2.5) |
| Smoking, yes | 20 (6.4) | 11 (5.6) | 5 (6.3) | 4 (11.4) |

Mean (standard deviation) is presented for continuous variables except for parity where median (interquartile range) is presented. For categorical variables n (%) is presented.

Overweight is defined as BMI 25.0–29.9 kg/m² and obesity as BMI ≥30.0 kg/m².

*Underweight (n = 2) is classified as normal weight.

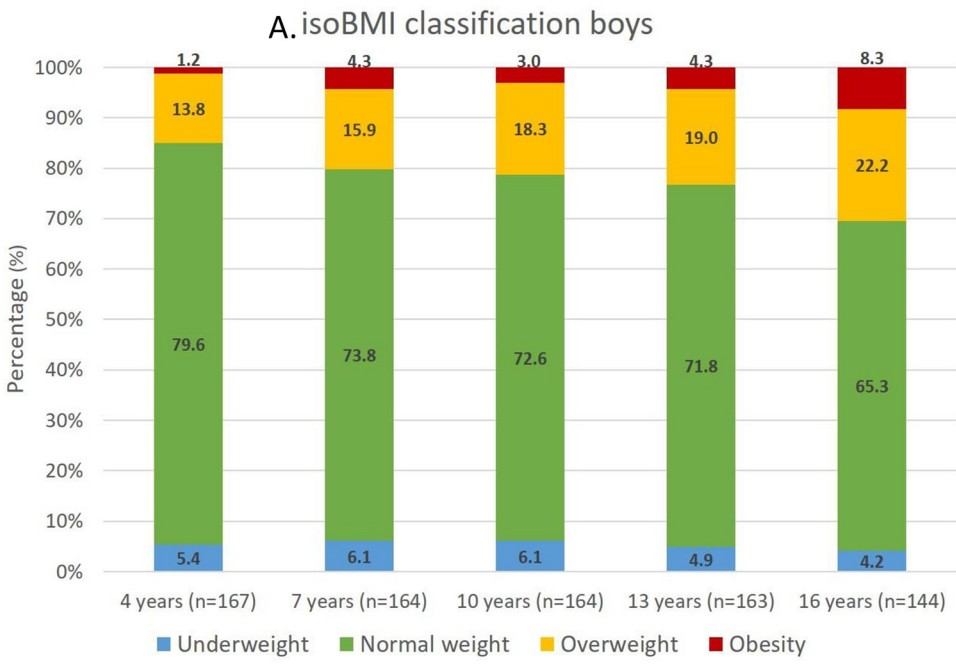

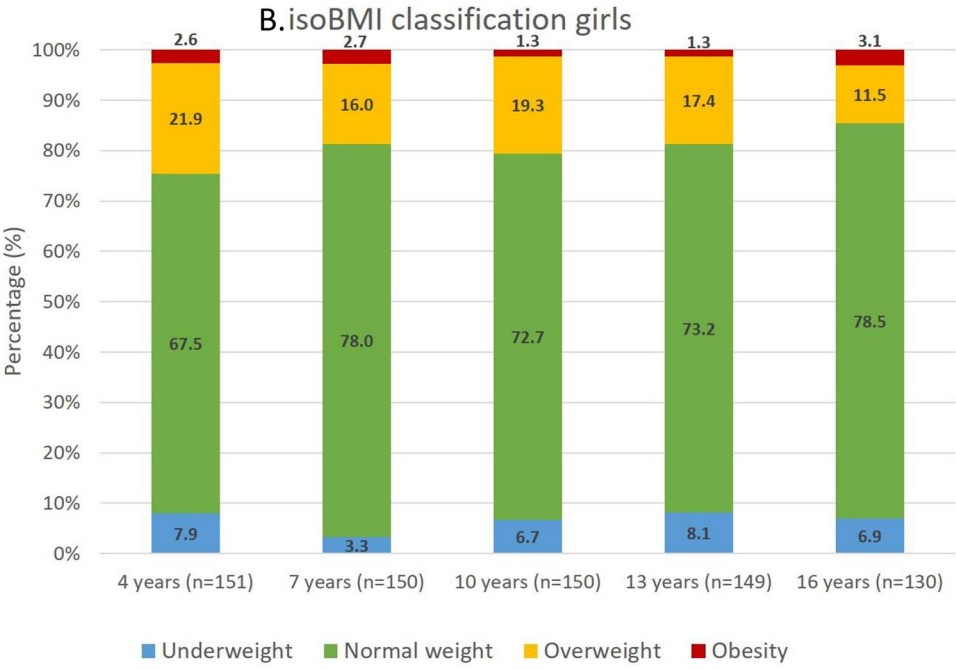

**Fig 2.** Distribution of body mass index (BMI) classes according to the isoBMI classification in children born 1999–2000 in the municipality Lidköping at different ages for boys (A) and girls (B) separately.

**Table 2. Association between maternal body mass index in early pregnancy and child weight at birth and 1 year, and child body mass index at 4, 7, 10, 13 and 16 years for the entire study population and for boys and girls separately.**

|  | Unadjusted model Beta (95% CI) | R-square | p-value | Adjusted model Beta (95% CI) | R-square | p-value |
|---|---|---|---|---|---|---|
| Maternal body mass index versus child weight in kg |  |  |  |  |  |  |
| Birth | 0.04 (0.02–0.05) | 0.06 | <0.001 | 0.03 (0.02–0.05) | 0.09 | <0.001 |
| Boys, n = 168 | 0.03 (0.01–0.05) | 0.06 | 0.006 | 0.03 (0.006–0.05) | 0.09 | 0.010 |
| Girls, n = 151 | 0.05 (0.02–0.07) | 0.05 | 0.003 | 0.04 (0.009–0.07) | 0.09 | 0.011 |
| 1 year age | 0.07 (0.03–0.10) | 0.05 | <0.001 | 0.06 (0.03–0.10) | 0.08 | <0.001 |
| Boys, n = 168 | 0.05 (0.01–0.09) | 0.04 | 0.007 | 0.05 (0.009–0.09) | 0.07 | 0.016 |
| Girls, n = 151 | 0.07 (0.004–0.14) | 0.04 | 0.039 | 0.07 (-0.002–0.13) | 0.10 | 0.057 |
| Maternal body mass index versus child body mass index |  |  |  |  |  |  |
| 4 years age | 0.10 (0.07–0.14) | 0.10 | <0.001 | 0.11 (0.07–0.14) | 0.11 | <0.001 |
| Boys, n = 167 | 0.10 (0.06–0.15) | 0.13 | <0.001 | 0.11 (0.06–0.15) | 0.14 | <0.001 |
| Girls, n = 151 | 0.12 (0.04–0.19) | 0.07 | 0.002 | 0.12 (0.05–0.19) | 0.10 | 0.001 |
| 7 years age | 0.22 (0.14–0.29) | 0.19 | <0.001 | 0.21 (0.14–0.29) | 0.21 | <0.001 |
| Boys, n = 164 | 0.23 (0.13–0.33) | 0.24 | <0.001 | 0.23 (0.13–0.32) | 0.26 | <0.001 |
| Girls, n = 150 | 0.19 (0.09–0.29) | 0.11 | <0.001 | 0.19 (0.09–0.29) | 0.14 | <0.001 |
| 10 years age | 0.29 (0.20–0.38) | 0.19 | <0.001 | 0.29 (0.21–0.38) | 0.21 | <0.001 |
| Boys, n = 164 | 0.32 (0.20–0.44) | 0.25 | <0.001 | 0.32 (0.20–0.43) | 0.27 | <0.001 |
| Girls, n = 150 | 0.24 (0.12–0.35) | 0.10 | <0.001 | 0.26 (0.15–0.38) | 0.13 | <0.001 |
| 13 years age | 0.34 (0.23–0.45) | 0.18 | <0.001 | 0.35 (0.24–0.46) | 0.19 | <0.001 |
| Boys, n = 163 | 0.39 (0.24–0.53) | 0.24 | <0.001 | 0.39 (0.23–0.54) | 0.25 | <0.001 |
| Girls, n = 149 | 0.27 (0.14–0.40) | 0.10 | <0.001 | 0.28 (0.14–0.42) | 0.11 | <0.001 |
| 16 years age | 0.47 (0.33–0.62) | 0.22 | <0.001 | 0.48 (0.33–0.63) | 0.24 | <0.001 |
| Boys, n = 144 | 0.55 (0.36–0.74) | 0.29 | <0.001 | 0.56 (0.37–0.76) | 0.31 | <0.001 |
| Girls, n = 130 | 0.27 (0.10–0.44) | 0.08 | 0.002 | 0.26 (0.07–0.44) | 0.11 | 0.006 |

The adjusted model was adjusted for maternal age, smoking status, and parity.

### Trajectories of BMI from 4 years of age to adolescence

Fig 3 shows the retrospective trajectories of BMI at 4, 7, 10 and 13 years of age, presented as percentage of children who were underweight, normal weight, overweight, or obese at these ages according to their weight class at 16 years age. None of the 16-year olds with underweight were either overweight or obese earlier in life (Fig 3A). Among 16-year olds with normal weight, the majority had a normal weight throughout childhood (Fig 3B). For 16-year-olds with overweight 31.9% were overweight or obese at the age of 4 (Fig 3C). Most adolescents with obesity had a normal weight at 4 years of age, but 37.6% were either overweight or obese at this age (Fig 3D).

## Discussion

### Main findings

This study showed that mBMI in early pregnancy was positively associated with child BMI throughout childhood to the age of 16 years and the correlation was strongest between mothers and 16-year-old sons. The risk of overweight or obesity at 16 years of age increased with increasing mBMI. In addition, most adolescents with normal weight had normal weight throughout childhood. We also found that the prevalence of overweight and obesity in 16-year-old children born 1999–2000 in the municipality Lidköping was 23.0%.

**Table 3. Relative-risk ratios (RRR) for 4, 7, 10, 13, and 16-year old children to be classified as underweight, overweight or obese as compared to normal weight, per 1 unit increase in maternal body mass index.**

| | Total | | Boys | | Girls | |
|---|---|---|---|---|---|---|
| | RRR (95% CI) | p-value | RRR (95% CI) | p-value | RRR (95% CI) | p-value |
| 4 years old | | | | | | |
| Unadjusted model | | | | | | |
| Normal weight | 1 (reference) | | 1 (reference) | | 1 (reference) | |
| Underweight | 0.96 (0.84–1.11) | 0.61 | 0.95 (0.78–1.15) | 0.60 | 0.99 (0.80–1.24) | 0.96 |
| Overweight | 1.17 (1.09–1.26) | <0.001 | 1.24 (1.13–1.36) | <0.001 | 1.15 (1.02–1.30) | 0.018 |
| Obesity | 1.34 (1.13–1.57) | 0.001 | 1.52 (1.08–2.14) | 0.017 | 1.43 (1.01–2.04) | 0.046 |
| Adjusted model | | | | | | |
| Normal weight | 1 (reference) | | 1 (reference) | | 1 (reference) | |
| Underweight | 0.97 (0.84–1.11)) | 0.63 | 0.96 (0.79–1.16) | 0.66 | 0.98 (0.78–1.23) | 0.86 |
| Overweight | 1.17 (1.09–1.26) | <0.001 | 1.24 (1.11–1.38) | <0.001 | 1.17 (1.03–1.34) | 0.016 |
| Obesity | 1.32 (1.15–1.52) | <0.001 | 1.36 (1.10–1.68) | 0.005 | 1.48 (0.93–2.36) | 0.097 |
| 7 years old | | | | | | |
| Unadjusted model | | | | | | |
| Normal weight | 1 (reference) | | 1 (reference) | | 1 (reference) | |
| Underweight | 0.95 (0.82–1.10) | 0.49 | 0.86 (0.71–1.04) | 0.11 | 1.13 (0.88–1.45) | 0.35 |
| Overweight | 1.19 (1.10–1.29) | <0.001 | 1.20 (1.09–1.32) | <0.001 | 1.18 (1.02–1.36) | 0.031 |
| Obesity | 1.41 (1.28–1.56) | <0.001 | 1.41 (1.24–1.60) | <0.001 | 1.49 (1.26–1.77) | <0.001 |
| Adjusted model | | | | | | |
| Normal weight | 1 (reference) | | 1 (reference) | | 1 (reference) | |
| Underweight | 0.96 (0.83–1.10) | 0.54 | 0.87 (0.73–1.03) | 0.097 | 1.09 (0.90–1.31) | 0.39 |
| Overweight | 1.19 (1.09–1.29) | <0.001 | 1.19 (1.08–1.32) | 0.001 | 1.16 (0.99–1.36) | 0.063 |
| Obesity | 1.40 (1.26–1.55) | <0.001 | 1.40 (1.24–1.58) | <0.001 | 1.57 (1.26–1.94) | <0.001 |
| 10 years old | | | | | | |
| Unadjusted model | | | | | | |
| Normal weight | 1 (reference) | | 1 (reference) | | 1 (reference) | |
| Underweight | 0.88 (0.77–1.01) | 0.068 | 0.87 (0.72–1.04) | 0.13 | 0.89 (0.73–1.10) | 0.29 |
| Overweight | 1.21 (1.12–1.31) | <0.001 | 1.24 (1.12–1.37) | <0.001 | 1.19 (1.04–1.37) | 0.012 |
| Obesity | 1.54 (1.27–1.87) | <0.001 | 1.74 (1.32–2.31) | <0.001 | 1.22 (0.99–1.49) | 0.063 |
| Adjusted model | | | | | | |
| Normal weight | 1 (reference) | | 1 (reference) | | 1 (reference) | |
| Underweight | 0.87 (0.76–1.00) | 0.056 | 0.86 (0.72–1.02) | 0.089 | 0.85 (0.69–1.05) | 0.13 |
| Overweight | 1.22 (1.12–1.32) | <0.001 | 1.24 (1.11–1.37) | <0.001 | 1.22 (1.05–1.42) | 0.008 |
| Obesity | 1.52 (1.25–1.85) | <0.001 | 1.65 (1.25–2.16) | <0.001 | 1.17 (1.01–1.36) | 0.040 |
| 13 years old | | | | | | |
| Unadjusted model | | | | | | |
| Normal weight | 1 (reference) | | 1 (reference) | | 1 (reference) | |
| Underweight | 0.85 (0.71–1.01) | 0.063 | 0.85 (0.64–1.14) | 0.28 | 0.85 (0.68–1.06) | 0.15 |
| Overweight | 1.19 (1.11–1.29) | <0.001 | 1.16 (1.06–1.26) | 0.002 | 1.29 (1.11–1.49) | 0.001 |
| Obesity | 1.54 (1.30–1.82) | <0.001 | 1.51 (1.27–1.80) | <0.001 | 1.47 (0.94–2.32) | 0.091 |
| Adjusted model | | | | | | |
| Normal weight | 1 (reference) | | 1 (reference) | | 1 (reference) | |
| Underweight | 0.84 (0.72–0.99) | 0.033 | 0.82 (0.63–1.06) | 0.13 | 0.85 (0.68–1.05) | 0.13 |
| Overweight | 1.19 (1.10–1.29) | <0.001 | 1.16 (1.05–1.28) | 0.003 | 1.30 (1.10–1.54) | 0.002 |
| Obesity | 1.54 (1.30–1.84) | <0.001 | 1.50 (1.27–1.77) | <0.001 | 1.56 (0.95–2.55) | 0.077 |
| 16 years old | | | | | | |

(*Continued*)

**Table 3.** (Continued)

|  | Total | | Boys | | Girls | |
| --- | --- | --- | --- | --- | --- | --- |
|  | **RRR (95% CI)** | **p-value** | **RRR (95% CI)** | **p-value** | **RRR (95% CI)** | **p-value** |
| Unadjusted model |  |  |  |  |  |  |
| Normal weight | 1 (reference) |  | 1 (reference) |  | 1 (reference) |  |
| Underweight | 0.89 (0.75–1.06) | 0.19 | 0.90 (0.71–1.14) | 0.37 | 0.88 (0.69–1.13) | 0.32 |
| Overweight | 1.21 (1.11–1.32) | <0.001 | 1.22 (1.09–1.36) | <0.001 | 1.15 (0.98–1.36) | 0.094 |
| Obesity | 1.42 (1.26–1.60) | <0.001 | 1.46 (1.27–1.67) | <0.001 | 1.26 (0.92–1.73) | 0.15 |
| Adjusted model |  |  |  |  |  |  |
| Normal weight | 1 (reference) |  | 1 (reference) |  | 1 (reference) |  |
| Underweight | 0.93 (0.79–1.08) | 0.34 | 0.95 (0.78–1.16) | 0.59 | 0.91 (0.73–1.14) | 0.43 |
| Overweight | 1.24 (1.13–1.35) | <0.001 | 1.25 (1.11–1.41) | <0.001 | 1.17 (0.99–1.38) | 0.065 |
| Obesity | 1.46 (1.29–1.65) | <0.001 | 1.52 (1.32–1.76) | <0.001 | 1.41 (0.98–2.03) | 0.068 |

The adjusted model was adjusted for maternal age, smoking status, and parity.

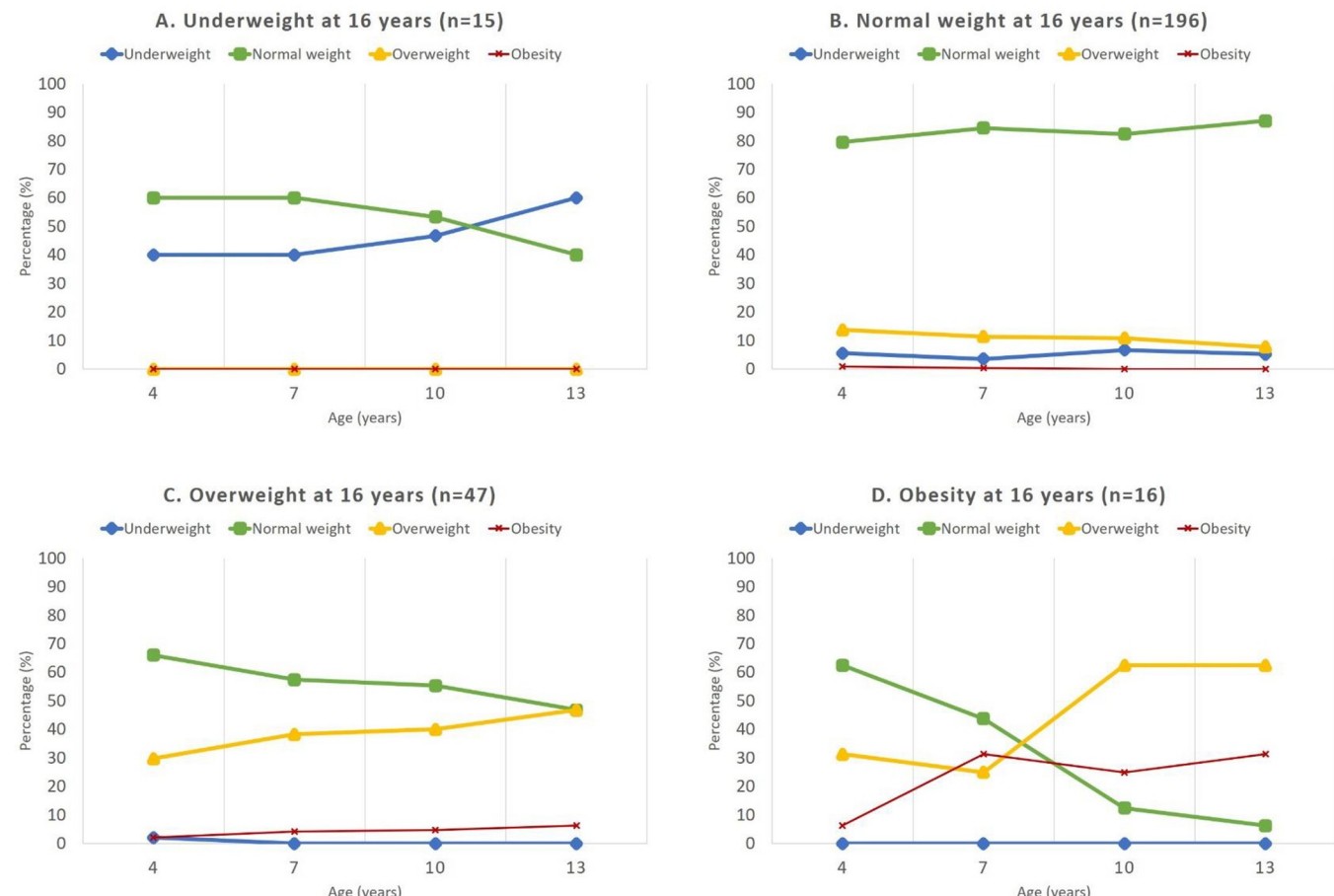

**Fig 3.** Development of weight class according to the isoBMI classification from early childhood to adolescence for children born 1999–2000 in Lidköping, divided into the weight class at 16 years; underweight (A), normal weight (B), overweight (C) and obesity (D).

## Strengths and limitations

The major strength of this study is the longitudinal design with maternal data from early pregnancy up to 16 years of age for the children. As the Swedish health care system offers free visits in both antenatal clinics and CHC centres we most likely included the majority of pregnant women and their children during 1999–2000. Data was available at several time points throughout childhood for most children, which enables estimations of prevalence of overweight and obesity at different ages. We used reliable measurements from antenatal clinics, CHC and school health care for calculating BMI/isoBMI and not self-reported body weight and height as in many other studies.

There are some limitations of the study, including a relatively small sample size. Further, the study population was homogeneous in terms of ethnicity, which limits the generalizability of the results. We had no information about maternal diabetes or gestational weight gain, both known to increase the risk of overweight and obesity in the offspring [23]. Also, information about other confounding factors, such as paternal BMI or socio-economic status was missing. Further, there were neither data on chronic illnesses or medication in the children nor data on lifestyle, which could have affected BMI progression. We used BMI as a proxy to assess overweight and obesity. This is a well-established method, but not as accurate as measuring body fat percentage since a higher amount of muscle mass could result in a higher BMI. Finally, BMI was recorded up to 15 weeks into pregnancy, which might not have been representative as pre-pregnancy BMI. However, Krukowski et al. [24] argued that most women are correctly classified in BMI classes based on a first trimester weight.

## Findings in relation to other studies

The prevalence of overweight or obesity at the age of 4 years (2003–2004) was 19.5% in our study compared to 11% in a report from Swedish Public Health Agency where almost all Swedish 4-year-old children were included 2018 [5]. A Swedish study from 2016 showed a national prevalence of overweight, including obesity, of 16.0% for boys and 12.8% for girls 12 years old [2]. In our study the corresponding prevalence for boys at the age of 13 years (2012–2014) was 23.3% and for girls 18.8%. This is higher than the national prevalence a few years later. In a recent report from Swedish Public Health Agency overweight and obesity were found in 21% of Swedish high school children [5] and in our study the prevalence was 23.0%. This difference might be a trend of decreasing prevalence of overweight and obesity during the later years. Our results could be explained by regional differences of overweight and obesity as our study population lived in a rural area where the frequency of overweight and obesity has been shown to be higher than in urban areas [25]. Of note, compared to other high-income Western countries the estimated percentage of children being obese is much lower in our studied population. The prevalence of childhood obesity in high-income Western countries 2016 was 16.8% for boys and 13.3% for girls aged 5–19 [26]. In our study the highest estimation of obesity was 8.3% for 16-year old boys.

The correlation between mBMI before or in early pregnancy and child BMI has been well established in earlier studies [10, 11]. Our result is in line with these studies and the correlation was positive for both sexes in all studied ages up to 16 years. The association is most likely due to a complex combination of genetic factors, environmental and behavioural factors during childhood where the lifestyle of the family plays a big part [27, 28]. The environment in utero also seems to have impact on a child's future growth and body composition. Pregnant women with overweight and particularly obesity transfer higher amounts of glucose, amino acids and free fatty acids through the placenta to the foetus compared to women with normal weight. This could result in permanent changes in control of appetite and energy metabolism in the

developing foetus, resulting in overweight and obesity later in life [29]. In later years, research has also focused on epigenetic processes and how the environment in utero modifies genes expression, for example through DNA-metylation [30]. New technologies are used in order to study genetic and *in utero* causes of overweight and obesity in offspring such as Mendelian randomization [31]. Also, genetic imprinting and DNA methylation might play important roles for future risk assessment of obesity [15]. It seems clear that having a mother with overweight or obesity increases the child's risk of having overweight or obesity but the exact mechanisms behind the causality are not yet understood. Our study gives no further explanation on the matter but confirms the association and emphasizes the need for further studies.

In our study the correlation between mBMI and child BMI grew stronger with increasing age in boys, but not in girls where the correlation overall was weaker. This suggests that maternal overweight and obesity may affect boys negatively to a higher degree than girls, which has been found in other studies [16, 25]. However, the opposite was found among 226 Swedish siblings [17], where a higher pre-pregnancy BMI was associated with increasing percentage fat mass in offspring in late adolescence, particular in daughters.

The sex difference in development of obesity depends on a multitude of factors including differential gene expression due to differences between the sexes in mRNA splicing, sex hormone effects, epigenetic mechanisms, mitochondrial and placental function [15]. In addition, the psychological pressure from peers and internet especially in teenage girls could influence the weight and bodily development [32]. The bodily changes during childhood and adolescence differ between individuals and occur at different points of time for instance occurrence of puberty [33]. These might be confounders in the analyses and could result in different outcomes in boys and girls and also explain deviating results in different studies.

A large German population-based study found that overweight and obesity manifested early in childhood and among overweight or obese adolescents, BMI had accelerated between 2 and 6 years of age [4]. In the current study we tracked BMI from 4 years of age and throughout childhood and found that 37.6% of 16-year olds with obesity had already had overweight or obesity early in childhood, indicating that early childhood may be a critical period for the development of sustained obesity. Epidemiological studies have previously shown that the risk of childhood obesity increases with an early "adiposity rebound", the phase in childhood during which BMI starts to increase [34].

## Conclusions

Our study confirms the correlation between maternal body mass index in early pregnancy and body mass index in children up to the age of 16 years. The prevalence of childhood overweight and obesity in the rural municipality we have studied was higher than previously reported nationally in Sweden. Our longitudinal study also confirms that obesity can be established in early childhood, which highlights the need for early intervention in the preventive work to combat childhood obesity, preferably starting at the antenatal clinic and follow-up in CHC centres and schools.

## Supporting information

**S1 Table. Relative-risk ratios (RRR) for 4, 7, 10, 13, and 16-year old children to be classified as underweight, overweight or obese as compared to normal weight, per 5 unit increase in maternal body mass index.**
(DOCX)

## Acknowledgments

We are deeply grateful to the nurses at the antenatal clinics, CHC centres and schools for their help during acquisition of data.

## Author Contributions

**Conceptualization:** Lisa Önnestam, Orsolya Haahr Vad, Tobias Andersson, Åsa Jolesjö, Jenny Sandegård, Kristina Bengtsson Boström.

**Data curation:** Lisa Önnestam, Orsolya Haahr Vad, Tobias Andersson, Åsa Jolesjö, Jenny Sandegård, Kristina Bengtsson Boström.

**Formal analysis:** Lisa Önnestam, Orsolya Haahr Vad, Tobias Andersson, Kristina Bengtsson Boström.

**Funding acquisition:** Lisa Önnestam, Orsolya Haahr Vad, Kristina Bengtsson Boström.

**Investigation:** Lisa Önnestam, Orsolya Haahr Vad, Tobias Andersson, Åsa Jolesjö, Jenny Sandegård.

**Methodology:** Lisa Önnestam, Orsolya Haahr Vad, Tobias Andersson, Åsa Jolesjö, Jenny Sandegård, Kristina Bengtsson Boström.

**Project administration:** Lisa Önnestam, Orsolya Haahr Vad, Tobias Andersson, Kristina Bengtsson Boström.

**Resources:** Lisa Önnestam, Orsolya Haahr Vad, Tobias Andersson.

**Supervision:** Tobias Andersson, Kristina Bengtsson Boström.

**Writing – original draft:** Lisa Önnestam.

**Writing – review & editing:** Lisa Önnestam, Orsolya Haahr Vad, Tobias Andersson, Åsa Jolesjö, Jenny Sandegård, Kristina Bengtsson Boström.

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
