## [Decision Letter · Decision Letter 0]

21 Jul 2022

PONE-D-22-15811Maternal body mass index in early pregnancy is associated with overweight and obesity in children up to 16 years of agePLOS ONE

Dear Dr. Boström,

Thank you for submitting your manuscript to PLOS ONE. After careful consideration, we feel that it has merit but does not fully meet PLOS ONE’s publication criteria as it currently stands. Therefore, we invite you to submit a revised version of the manuscript that addresses the points raised during the review process.

We look forward to receiving your revised manuscript.

Kind regards,

David Meyre

Academic Editor

PLOS ONE

Journal Requirements:

We are deeply grateful to the nurses at the antenatal clinics, CHC centres and schools for their help during acquisition of data. The work was supported by the Skaraborg research and development council (grant: VGFOUSKB-869761).

 The work was supported by the Skaraborg research and development council (grant: VGFOUSKB-869761).

6. PLOS requires an ORCID iD for the corresponding author in Editorial Manager on papers submitted after December 6th, 2016. Please ensure that you have an ORCID iD and that it is validated in Editorial Manager. To do this, go to ‘Update my Information’ (in the upper left-hand corner of the main menu), and click on the Fetch/Validate link next to the ORCID field. This will take you to the ORCID site and allow you to create a new iD or authenticate a pre-existing iD in Editorial Manager. Please see the following video for instructions on linking an ORCID iD to your Editorial Manager account: https://www.youtube.com/watch?v=_xcclfuvtxQ.

7. We note that you have included the phrase “data not shown” in your manuscript. Unfortunately, this does not meet our data sharing requirements. PLOS does not permit references to inaccessible data. We require that authors provide all relevant data within the paper, Supporting Information files, or in an acceptable, public repository. Please add a citation to support this phrase or upload the data that corresponds with these findings to a stable repository (such as Figshare or Dryad) and provide and URLs, DOIs, or accession numbers that may be used to access these data. Or, if the data are not a core part of the research being presented in your study, we ask that you remove the phrase that refers to these data.

Reviewers' comments:

Reviewer's Responses to Questions

**Comments to the Author**

1. Is the manuscript technically sound, and do the data support the conclusions?

Reviewer #1: Yes

Reviewer #2: Yes

2. Has the statistical analysis been performed appropriately and rigorously? 

Reviewer #1: Yes

Reviewer #2: Yes

3. Have the authors made all data underlying the findings in their manuscript fully available?

Reviewer #1: No

Reviewer #2: No

4. Is the manuscript presented in an intelligible fashion and written in standard English?

Reviewer #1: Yes

Reviewer #2: Yes

5. Review Comments to the Author

Reviewer #1: This study by Önnestam et al. explores the longitudinal association between maternal BMI in early pregnancy and body mass index in children up to the age of 16 years in a regional Swedish population. Overall, this prospective study is well designed, method is appropriate, and the results are clear and well organized. The manuscript is interesting and has strengths, including repeated measurements over time. Some comments and suggestions are as follows:

1. In the last paragraph of the introduction, one of the objectives stated includes exploration of the correlation between maternal BMI and child BMI in gender-specific subgroups. However, there is no context provided as to why these subgroups were investigated. It would be helpful to provide context for this objective by adding a few lines in an earlier paragraph as to why analyzing BMI in boys and girls separately is important based on what is known from literature about sex-based differences in BMI.

2. Overall, the methodology is clearly explained. The results are also displayed clearly and sequentially. Information in the tables is also presented clearly.

3. In the results section, while the association between maternal BMI and child weight/BMI has been explored at all defined ages (i.e. 4, 7, 10, 13, 16), the association between maternal BMI and risk of child overweight or obesity has only been assessed at the age of 16. It may be interesting to also see association between maternal BMI and child obesity/overweight risk at the other specified ages as well. In addition, the authors may consider adding analysis that investigates maternal BMI as a categorical variable based on weight status (normal weight, overweight, or obese) in association with risk of child overweight/obesity (currently maternal BMI has only been analyzed as a continuous variable). This may provide clinically practical results for groups of pregnant women who fall into each weight category in addition to looking at risk associated with unit changes in maternal BMI.

4. In the discussion section, the authors should consider expanding discussion on changes in BMI associated with age due to normal growth and development and how that may be a confounder when analyzing changes in obesity/overweight trajectory, especially given that this is a pediatric sample. Additionally, the authors may consider discussing potential explanations for the difference in results noted between boys and girls (i.e. higher prevalence of overweight and obesity in boys and stronger correlation with mBMI) - for instance discussion of sex specific risk factors.

5. The manuscript requires minor grammatical revisions.

Reviewer #2: This study investigates the associations between maternal BMI in early pregnancy and the risk of child overweight/obesity at age 16. It is well written, clear and easy to read. However, the patterns of associations between pre-pregnancy BMI and offspring overweight has been extensively studied in the literature. Therefore, I regret to acknowledge that, despite the quality of the manuscript, this study does not add novelty to what is already know in the topic.

There may be some geographical specificities but they are not investigated in sufficient details. Moreover, the number of potential confounder taken into account is very low and makes the results subject to residual confounding.

Another objective of the paper is to describe tracking from 4 to 16 years, but the articulation between both objectives is not obvious.

I would also have additional minor comments and suggestions for the authors as follows:

- Models used are logistic models, so results should be odds ratio instead of Relative risks

- Why using 5-point maternal BMI increase, it’s a very important difference which artificially leads to extremely high ORs

- The § dedicated to the description of clinical characteristics is rather long and could be better organized. There seems to be interesting and sex-specific evolutions of prevalence, they could be described as such.

- In the results rather display and describe the adjusted rather than the unadjusted results (even if important confounders are lacking)

- Table 2: provide the numbers. And precise whether the numbers are all the same over the different ages (no missing value)?

- Some references are missing here and there in the discussion. For example, for the statement “we had no information about maternal diabetes or gestational weight gain, both known to increase the risk of overweight ad obesity in the offspring”; or “the prevalence of childhood obesity in high-income western countries…”

- There is a debate in the scientific community, especially due to recent results from mendelian randomisation studies, regarding the intrauterine programming of obesity hypothesis and the role of maternal pre-pregnancy obesity in this programming; this should be discussed in more details.

6. PLOS authors have the option to publish the peer review history of their article (what does this mean?). If published, this will include your full peer review and any attached files.

Reviewer #1: No

Reviewer #2: No

---

## [Author Response · Author response to Decision Letter 0]

13 Sep 2022

PLOSONE-D-22-15811

Maternal body mass index in early pregnancy is associated with overweight and obesity in children up to 16 years of age

Answer: We would like to thank the Editor and reviewers for encouraging and constructive questions, comments and suggestions. Please find below our answers point by point. 

Journal Requirements:

Answer: We have amended the manuscript in order to comply with the PLOS ONE style requirements. In addition we have updated the web-links of 2 references (#5 and 20). In figure 2 have changed the orders of the panels so that boys are presented first in congruence with data presentation in the main text. 

Answer: The study was approved by the Regional Ethics Review Board in Gothenburg (reference 935-18). We have not retrieved informed consent from the studied individuals since we are reporting results from retrospective data from registers. The need for consent was waived by the ethics committee.

Answer: We have checked the grant number and added information of the grants role in the project. Please see below, point 4. 

We are deeply grateful to the nurses at the antenatal clinics, CHC centres and schools for their help during acquisition of data. The work was supported by the Skaraborg research and development council (grant: VGFOUSKB-869761).

The work was supported by the Skaraborg research and development council (grant: VGFOUSKB-869761).

Answer: We have now up-dated the funding information as follows and removed it from the manuscript: 

The work was supported by the Skaraborg research and development council, Grant number: VGFOUSKB-869761. Web–site: http://www.vgregion.se/fouskaraborg. The grant was awarded to the first author, LÖ in 2018. The funders had no role in study design, data collection and analysis, decision to publish, or preparation of the manuscript.

We have included the amended statement in the Cover letter. Thank you very much for your help in changing the text in the online submission.

Answer: Data cannot be shared publicly because of ethical and legal restrictions from the Swedish authorities as the data contain sensitive and potentially identifying patient information. Data may however be available for researchers who meet the criteria for access to confidential data, upon reasonable request to the authors and with permission from the Swedish Ethical Review Authority (https://etikprovningsmyndigheten.se, registrator@etikprovning.se).

6. PLOS requires an ORCID iD for the corresponding author in Editorial Manager on papers submitted after December 6th, 2016. Please ensure that you have an ORCID iD and that it is validated in Editorial Manager. To do this, go to ‘Update my Information’ (in the upper left-hand corner of the main menu), and click on the Fetch/Validate link next to the ORCID field. This will take you to the ORCID site and allow you to create a new iD or authenticate a pre-existing iD in Editorial Manager. Please see the following video for instructions on linking an ORCID iD to your Editorial Manager account: https://www.youtube.com/watch?v=_xcclfuvtxQ.

Answer: The corresponding author (KBB) has up-dated her ORCID id. 

7. We note that you have included the phrase “data not shown” in your manuscript. Unfortunately, this does not meet our data sharing requirements. PLOS does not permit references to inaccessible data. We require that authors provide all relevant data within the paper, Supporting Information files, or in an acceptable, public repository. Please add a citation to support this phrase or upload the data that corresponds with these findings to a stable repository (such as Figshare or Dryad) and provide and URLs, DOIs, or accession numbers that may be used to access these data. Or, if the data are not a core part of the research being presented in your study, we ask that you remove the phrase that refers to these data.

Answer: We have included data formerly “not shown” in Table 2, as well as adjusted data as requested by the reviewers.

Reviewers' comments:

Reviewer's Responses to Questions

Comments to the Author

1. Is the manuscript technically sound, and do the data support the conclusions?

Reviewer #1: Yes

Reviewer #2: Yes

Answer: Thank you!

2. Has the statistical analysis been performed appropriately and rigorously?

Reviewer #1: Yes

Reviewer #2: Yes

Answer: Thank you!

3. Have the authors made all data underlying the findings in their manuscript fully available?

Reviewer #1: No

Reviewer #2: No

Answer: Please read our answers to items #5 and #7 under subheading Journal Requirements.

4. Is the manuscript presented in an intelligible fashion and written in standard English?

Reviewer #1: Yes

Reviewer #2: Yes

Answer: Thank you!

5. Review Comments to the Author

Reviewer #1: This study by Önnestam et al. explores the longitudinal association between maternal BMI in early pregnancy and body mass index in children up to the age of 16 years in a regional Swedish population. Overall, this prospective study is well designed, method is appropriate, and the results are clear and well organized. The manuscript is interesting and has strengths, including repeated measurements over time. Some comments and suggestions are as follows:

Answer: Thank you for this encouraging comment!

Reviewer #1: 1. In the last paragraph of the introduction, one of the objectives stated includes exploration of the correlation between maternal BMI and child BMI in gender-specific subgroups. However, there is no context provided as to why these subgroups were investigated. It would be helpful to provide context for this objective by adding a few lines in an earlier paragraph as to why analyzing BMI in boys and girls separately is important based on what is known from literature about sex-based differences in BMI.

Answer: Thank you very much for this suggestion. We have now added a few lines (Lines 75-79) in the Introduction (manuscript with track changes) to present the rationale for studying BMI in boys and girls separately. “The impact of adverse uterine environment factors on metabolic traits later in life differs between the sexes as described in a newly published review [15]. However, while some observational studies indicate that boys may have a higher risk of developing obesity following maternal over-nutrition [16] the opposite has also been found [17]”.

Reviewer #1: 2. Overall, the methodology is clearly explained. The results are also displayed clearly and sequentially. Information in the tables is also presented clearly.

Answer: Thank you very much.

Reviewer #1: 3. In the results section, while the association between maternal BMI and child weight/BMI has been explored at all defined ages (i.e. 4, 7, 10, 13, 16), the association between maternal BMI and risk of child overweight or obesity has only been assessed at the age of 16. It may be interesting to also see association between maternal BMI and child obesity/overweight risk at the other specified ages as well. In addition, the authors may consider adding analysis that investigates maternal BMI as a categorical variable based on weight status (normal weight, overweight, or obese) in association with risk of child overweight/obesity (currently maternal BMI has only been analyzed as a continuous variable). This may provide clinically practical results for groups of pregnant women who fall into each weight category in addition to looking at risk associated with unit changes in maternal BMI.

Answer: Thank you for this important suggestion. We have expanded table 3 with the association between maternal BMI and child obesity/overweight risk at the other specified ages as well per 1 unit increase in maternal body mass index. We have also revised the text above Table 3. Correspondingly we have altered the text in Statistical methods (Lines 138-142). As our data sample size is limited, we have chosen to refrain from categorizing data on maternal BMI as this would reduce the statistical power in our analyses considerably. However, the presented RRRs (supplementary S1 Table) per 5 units increase in maternal BMI corresponds to an increase in maternal weight status from normal weight to overweight to obesity.

Reviewer #1: 4. In the discussion section, the authors should consider expanding discussion on changes in BMI associated with age due to normal growth and development and how that may be a confounder when analyzing changes in obesity/overweight trajectory, especially given that this is a pediatric sample. Additionally, the authors may consider discussing potential explanations for the difference in results noted between boys and girls (i.e. higher prevalence of overweight and obesity in boys and stronger correlation with mBMI) - for instance discussion of sex specific risk factors.

Answer. We have added text in the Discussion (lines 302-310): “The sex difference in development of obesity depends on a multitude of factors including differential gene expression due to differences between the sexes in mRNA splicing, sex hormone effects, epigenetic mechanisms, mitochondrial and placental function [15]. In addition, the psychological pressure from peers and internet especially in teenage girls could influence the weight and bodily development [32]. The bodily changes during childhood and adolescence differ between individuals and occur at different points of time for instance occurrence of puberty [33]. These might be confounders in the analyses and could result in different outcomes in boys and girls and also explain deviating results in different studies”.

Reviewer #1: 5. The manuscript requires minor grammatical revisions.

Answer: The manuscript has been subjected to linguistic revision and amended according to the suggestions of the linguistic reviewer. 

Reviewer #2: This study investigates the associations between maternal BMI in early pregnancy and the risk of child overweight/obesity at age 16. It is well written, clear and easy to read. However, the patterns of associations between pre-pregnancy BMI and offspring overweight has been extensively studied in the literature. Therefore, I regret to acknowledge that, despite the quality of the manuscript, this study does not add novelty to what is already know in the topic.

There may be some geographical specificities but they are not investigated in sufficient details. Moreover, the number of potential confounder taken into account is very low and makes the results subject to residual confounding.

Another objective of the paper is to describe tracking from 4 to 16 years, but the articulation between both objectives is not obvious

Answer: Thank you for addressing these important points. We agree that there are former studies that report the association between mother and child weight, as we also acknowledge in Introduction (lines 68-71, in Revised manuscript with track changes), “Numerous studies have shown an association…..” and in the Discussion (276-277), “The correlation between mBMI before or in early pregnancy and child BMI ……”

There are fewer studies, though, with a long time follow up and with consecutive measurements over time. Therefore, we think that this study contributes to the knowledge of development of overweight and obesity in children and specifically the difference between boys and girls in this respect. 

Of course we wished that potential confounders could have been included in the analyses, but the clinical health records we used are not originally intended for research and did not supply such information. We have described this limitation of the study in the Discussion, subheading Strengths and limitations, lines 248-251) “…information about other confounding factors, such as paternal BMI or socio-economic status was missing. Further, there were neither data on chronic illnesses or medication in the children nor data on lifestyle, which could have affected BMI progression”.

We describe that accelerating BMI in childhood has been shown to be associated to obesity in adolescence (lines 57-58, with references #4 and 6). Our data also allowed us to study this important and modifiable risk for overweight and obesity. To emphasize this we have altered the text about the rationale for tracking BMI of the children from 4 to 16 years of age. 

In Introduction (lines 58-59), we added “Thus, increasing BMI in children predicts overweight and obesity in adolescence and is therefore a target for preventive measures.” and we also altered the Conclusion, (lines 326-328) “ …which highlights the need for early intervention in the preventive work to combat childhood obesity, preferably starting at the antenatal clinic and follow-up in CHC centers and schools.”

Reviewer #2: I would also have additional minor comments and suggestions for the authors as follows:

Reviewer #2: - Models used are logistic models, so results should be odds ratio instead of Relative risks

Answer: Thank you for the comment. We have performed the multinominal logistic regressions using the Stata command mlogit with the rrr option. According to the Stata reference manual this reports the estimated coefficients transformed to relative risk ratios and not odds ratios. Please see example 3 with the accompanying technical note regarding this subject in the Stata online reference manual (pages 7-8):

https://www.stata.com/manuals/rmlogit.pdf

Reviewer #2: - Why using 5-point maternal BMI increase, it’s a very important difference which artificially leads to extremely high ORs

Answer: Thank you for the comment. The rationale to describe RRRs per 5-point maternal BMI increase was to correlate the results to the 5 unit increase in BMI classification from normal weight (BMI 20-25) to overweight (BMI 25-30) and from overweight to obesity (BMI > 30). The presented RRRs associated with a 5-point maternal BMI increase are equal to the RRRs of a 1-unit maternal BMI increase to the power of 5. The reported p-values are the same using either RRRs of a 1- or 5-point maternal BMI increase. However, we agree that a 5-point maternal BMI increase is quite substantial and have accordingly changed the text in the methods and results section and Table 3 to reflect RRRs of 1-point maternal BMI increase. In addition, we have provided results expressed as RRRs of 5-point maternal BMI increase in supplementary S1 Table.

Reviewer #2: - The § dedicated to the description of clinical characteristics is rather long and could be better organized. There seems to be interesting and sex-specific evolutions of prevalence, they could be described as such.

Answer: Thank you for this suggestion, we have shortened the text accordingly (Clinical characteristics of the mothers) as the data are shown in Figure1. In Clinical characteristics of the children we found that most data are not obvious from Figure 2 so we did not shorten the text. In Figure 2 we reversed the panels for girls and boys to be congruent with the order in the text. 

Reviewer #2: - In the results rather display and describe the adjusted rather than the unadjusted results (even if important confounders are lacking)

Answer: We have now expanded Table 2 to include both adjusted and unadjusted data. In the abstract (lines 34-36) and the result section (lines 190-192, 203) we now describe results from the adjusted analyses.

Reviewer #2: - Table 2: provide the numbers. And precise whether the numbers are all the same over the different ages (no missing value)?

Answer: We have provided the number of individuals (boys and girls separately) that contributed with data at different ages, please find it in Table 2. 

Reviewer #2: - Some references are missing here and there in the discussion. For example, for the statement “we had no information about maternal diabetes or gestational weight gain, both known to increase the risk of overweight ad obesity in the offspring”; or “the prevalence of childhood obesity in high-income western countries…”

Answer: Thank you, we have added a reference; a review from 2017, reference #23 (Agarwal P et al 2017) to the statements above according to the reviewers suggestion and moved reference #26 (former #24) to the end of the next sentence in the text to clarify the statement. 

Reviewer #2: - There is a debate in the scientific community, especially due to recent results from mendelian randomisation studies, regarding the intrauterine programming of obesity hypothesis and the role of maternal pre-pregnancy obesity in this programming; this should be discussed in more details.

Answer: Thank you for this suggestion. We have added a few lines regarding this matter in the Discussion part (lines 288-291). “New technologies are used in order to study genetic and in utero causes of overweight and obesity in offspring such as Mendelian randomization [31]. Also, genetic imprinting and DNA methylation might play important roles for future risk assessment of obesity [15]”.

---

## [Editor Report · Decision Letter 1]

19 Sep 2022

Maternal body mass index in early pregnancy is associated with overweight and obesity in children up to 16 years of age

PONE-D-22-15811R1

Dear Dr. Boström,

We’re pleased to inform you that your manuscript has been judged scientifically suitable for publication and will be formally accepted for publication once it meets all outstanding technical requirements.

Kind regards,

David Meyre

Academic Editor

PLOS ONE
---

## [Editor Report · Acceptance letter]

28 Sep 2022

PONE-D-22-15811R1 

Maternal body mass index in early pregnancy is associated with overweight and obesity in children up to 16 years of age 

Dear Dr. Bengtsson Boström:

I'm pleased to inform you that your manuscript has been deemed suitable for publication in PLOS ONE. Congratulations! Your manuscript is now with our production department. 

Kind regards, 

on behalf of

Dr. David Meyre 

Academic Editor

PLOS ONE